# Finite Element Modeling of Hot Rolling of 1075 Carbon Steel Process with Variable Cross Section

**DOI:** 10.3390/ma16010002

**Published:** 2022-12-20

**Authors:** Karina Montemayor-de la Garza, Patricia del Carmen Zambrano-Robledo, Oscar Jesus Zapata-Hernandez, Luis Adolfo Leduc-Lezama

**Affiliations:** Universidad Autonoma de Nuevo Leon, FIME-Centro de Investigación e Innovacion en Ingenieria Aeronautica (CIIIA), Av. Universidad S/N, Ciudad Universitaria, San Nicolás de los Garza 66455, Mexico

**Keywords:** hot rolling, railway, material flow, 1075 carbon steel, finite element method

## Abstract

Currently, it is common to use steel poles for applications in livestock and agriculture. In this work, finite element analysis of five hot rolling passes for the manufacture of farm poles using 1075 carbon steels from recycled railway material was developed. The steel industry in Mexico imports products from other countries or from companies specialized in metallurgy at an excessive cost. To be more competitive and save costs, companies seek the reutilization of existing resources such as the railway 1075 steel, which has good mechanical properties. SFTC DEFORM-3D software was used to model five hot rolling passes considering a variable cross section railway profile. The effect of rolling speed and temperature were considered to analyze flow behavior. Rolling loads were also determined.

## 1. Introduction

The finite element method (FEM) is a valuable tool for the evaluation of hot rolling. It considers three-dimensional geometries. Coupled thermo-elastoplastic behavior can be considered for modeling, and it allows evaluation of the effect of roll geometry strip profile, temperature, friction between the work rolls and the slab, thickness reduction, and speed of rotation of the work rolls in the process [1]. Simulating a complete hot rolling mill is a challenge. However, some benefits include estimation of final length of the manufactured element and its variables: the geometry of the cross section on each pass, the effective stress, effective plastic strain, and rolling power [2]. A series of publications have been documented where FEM simulation tools were used for rolling processes, combining mathematical modeling methods with laboratory tests considering important variables such as coefficient of friction, temperature, microstructure evolution, and shape of the laminated product [3].

There are works where finite element packages such as DEFORM have been used for analysis of surface defects in continuous casting slabs [4]. In [5], the authors used a plastic work approach to analyze hot rolling to estimate mechanical behavior due to modification of geometries during the rolling process in hot bars. In other works, superficial defects of products generated by hot rolling passes are evaluated considering different temperatures. Temperature validation using pyrometer measurements were also developed [6]. In [7], DEFORM-3D was used to simulate flow of material. The results of the simulations of metal formation agreed with cross sectional micrographs obtained from an experimental mill.

There are also works that focused on the microstructural behavior of steel during hot rolling. These models used temperature or pressure data as input for calculations [8]. Structural and temperature predictions were made to determine the average stress required to deform steel in a hot rolling process and results were compared with values calculated from the assumption of an adhesive friction model. However, the model did not consider the accumulation of deformation between rolling passes [9]. Other works consider phase transformation phenomena produced in the hot rolling process, such as the case of the transformation of austenite to ferrite as shown in [10] using macroscopic kinetics modeling, a thermodynamics model, and phase field. In reference [11], rod rolling passes were studied using MSC Marc Auto forge software. In [12] linear regression models were used to predict rolling force, wear, and thermal behavior of rolls. Results were validated with experimental data. In references [13,14], a thermomechanical characterization of quality of R260 steel, pearlitic steel 0.7% C, commonly used in rail rolling, was developed to study a hot rolling process. The results of maximum deformation and stresses generated during the rolling were shown. In [15] Cold and hot rolling of a rectangular cross section were studied, determining the equivalent stresses generated during the rolling pass. In [16], the authors proposed a combined rigid-viscoplastic Eulerian model to obtain the deformation of the steel plates during rolling, validating results with real data of this process. The use of neural network (ANN) methodologies in hot strip rolling mills using constant volume elements has also been published [17]. Deformation can be extraordinarily complex when using multi-pass rolling with variable cross sections, and temperature distribution is also not homogenous. There are some works that used dislocation theory equations to characterize the behavior of the material during complex deformation [18,19,20]. Other works expose the challenges of simulation technology that exhibit gaps between material data and computational technology of practical applications in the automotive world [21]. In [22] FEM modeling of hot rolling of tube rails considering three-dimensional coupled thermomechanical analysis was developed to study the continuous rolling process of several passes. The stress, strain, temperature, and rolling forces were calculated. In [23,24], the Taguchi optimization technique was used to predict the best results based on inputs such as roll diameter and friction value during rolling of AISI 1016 steel. The model provided diameters and lengths of the rolling, distance between rolling supports, and the tolerances that the steel can maintain.

Many of the mechanisms that govern the hot rolling process are still not fully understood, and there is a need to provide engineers with comprehensive tools that enable them to design rolls correctly the first time, thus reducing the number of design trials of rolls, the amount of material waste, and the cost of tooling, therefore reducing the time roll designers can spend on engineering, as well as improving confidence in the manufacturing process and end product quality [2,25,26].

In this work, a 1075 steel was analyzed in a hot rolling process. The geometry of steel was obtained from a rail with an I cross section. This section was cut in half to obtain the initial geometry for the passes. The chemical composition of the steel is shown in Table 1.

## 2. Mathematical Modeling

Some of the main challenges when modeling hot rolling processes are the inclusion of real variables in the mathematical model, that is, care must be taken so that the problem does not become so complex that results do not converge on a feasible solution. The success or failure of the model will depend on the precision with which this numerical representation is constructed.

The relationship between the process parameters, the mechanical properties of the material, and the force necessary for rolling has been described by Roberts and Leonard [27]. Models for hot rolling are based on the general relationship stated in plasticity theory, as shown in Equation (1):(1)$${\bar{\sigma }}_{t}=\frac{2}{\sqrt{3}}\sigma_{0}F$$
where $\sigma_{0}$ is the elastic limit, ${\bar{\sigma }}_{t}$ is the stress needed for rolling, and *F* is a function that calculates the influence of the friction coefficient *µ* and a form factor Δ, associated with the deformation zone, that is:*F* = *F*(*µ*,Δ)(2)

The model takes the separation force of the rollings *P*, which depends on the rolling length *L*, the average effective rolling resistance $\bar{k}$, and a geometric factor *Q*, as shown in Equation (3):(3)$$P=\bar{k}LQ$$

The average effective resistance to deformation is calculated from the strain *ε*, the strain rate $\overset{`}{\varepsilon }$, the absolute temperature *T*, and a set of coefficients ($\sigma_{0},\;K,\;n,\;m,\;C$) that depend on the material, as shown in Equation (4).
(4)$$\bar{k}=\frac{2}{\sqrt{3}}{\left(\sigma_{0}+K\left[\frac{2}{\sqrt{3}}\left(\varepsilon -\varepsilon_{0}\right)\right]\right)}^{n}{\left(\frac{2}{\sqrt{3}}\overset{`}{\varepsilon }\right)}^{m}\exp\left(-CT\right)$$

Figure 1 shows a stress–strain curve. For the 1075 steel, temperature and strain rate dependence curves are considered.

## 3. Numerical Modeling

In this work, SFTC DEFORM-3D software was used. A rail profile with an I section was cut into 2 parts, resulting in T shape and inverse T shape geometries, respectively. This provided the initial geometry for the first pass. Geometry and roll design were modeled as full 3D bodies.

Figure 2 shows the I shape rail modeled in CAD software. On the right, the T shape profile is shown. Dimensions are in millimeters. The rail was cut into two halves along the length, keeping the area of the two sections the same. In this work, only the top half of the rail was used.

The bottom and top rolls used for the first three rolling passes are shown in Figure 3, Figure 4 shows details of their angles and dimensions, as well as the silhouette of the rolling. The diameter is 355 mm.

The rolls used for the last 2 rolling passes are shown in Figure 5, in which Figure 5B shows details of their dimensions. The diameter is 304.8 mm.

Table 2 shows elastic properties of 1075 steel.

Thermal properties were considered dependent on temperature, and they are shown in Figure 6a,b.

Boundary conditions for hot rolling process were the following:Ambient temperature constant was 25 °C.Heat transfer coefficient was assumed as 20 W/m^2^ °C. This value considers cooling in air.Friction coefficient was 0.3.

Modeling assumptions:Isotropic behavior.Roll geometries were rigid, therefore the only mesh needed was the resolution of the STL file to improve surface contact conditions.

The hot rolling evaluation is summarized in the following data: (A)Upper rail part (T shape).(B)Five rolling passes.

Rolling temperature with three different settings:(A)1050 °C.(B)1150 °C.(C)1250 °C.

Rolling speeds with three different settings:(A)3.1 RPM.(B)6.2 RPM.(C)12.4 RPM.

Figure 7 shows the flowchart of the modeling of the hot rolling process.

The finite element mesh of the upper rail is shown in Figure 8. The number of elements was about 50,000. Adaptive mesh refinement was activated during simulation of each pass.

## 4. Results

The upper part of the rail was selected, and comparison was made of the first five passes with different conditions. Two rollings were considered for compressing the rail, there was a guide for the rail, and a thrust element controlled the speed of the rail.

The information obtained by the finite element analysis during the simulations is shown in Table 3.

Figure 9 shows the evolution for each rolling pass starting from the initial T shape geometry. In the first pass, initial geometry was rotated 180 degrees.

Table 4 shows the rolling loads and torques obtained during the passes. For all passes, for a certain angular velocity of rolls, as temperature increases, the rolling load decreases. When increasing rolling speed, sensitivity of the material to strain rate is activated and therefore rolling load increases. Pass 2 shows the same trend seen in the first pass where there was greater torque and stress in the pieces rolled at a lower temperature. It should be noted that unlike the first pass, this run showed minimal differences between the times of rolling at different speeds, which could be because the second rolling pass was the one that gave the rail the greatest modification, reducing its height by more than 40%.


**Pass 1:**


Figure 10 shows the results obtained from the first rolling pass considering three values of temperature and rolling speed. It can be observed that heterogeneity of deformation is different when increasing the temperature. This may be due to the nature of different mechanical properties due to temperature changes. It can also be noticed that contact area does not change when increasing rolling speed. Additionally, notice that curvature of the billet changes as temperature increases. This usually happens in practice, causing material to become stuck due to exceedingly high curvatures. High costs can be generated when “flattening” the workpiece to enter the next pass.

Figure 11 shows the effective strain comparison between results from the first rolling pass. As roll speed increases, more highly concentrated strain regions are detected in the thinner region of the geometry.

Figure 12 shows the temperature distribution after the first pass. These heterogenous temperature values are used for the next passes. The same color scale is used only for comparison purposes.

Once the different items of the rail have been analyzed for the first pass, a similar evaluation is conducted for the rest of the rolling passes (Figure 13 and Figure 14).


**Pass 2:**


Effective strain of the second pass is observed in the Figure 13 and effective stress of the second pass is observed in the Figure 14.

For the temperature graph shown in Figure 15, in the evaluation of the second pass of the upper rail, homogeneous temperature conditions in the piece can be mentioned, where no hot spots are observed derived from the concentration of temperature in specific points.


**Pass 3：**


Figure 16 shows the stress vs. temperature relationship consistent with what was expected and pass 2 shows the greatest impact on the modification of the geometry in relation to the five documented passes.

In terms of effective stress, (Figure 17), stress concentrations can be observed in at least six of the nine runs shown, having effective stress points in specific areas of the rail, and runs of 1150 °C at 6.2 RPM, 1150 °C at 12.4 RPM, and 1050 °C at 12.4 RPM are the three runs that show the best results. It stands out that these specific points of stress are not found in the area of deformation of rollers, and additional studies will be carried out for these cases to identify areas of opportunity in the proposed methodology.

Figure 18 shows the temperature graph obtained for the tirth pass for all the different scenarios of this work. 


**Pass 4:**


Figure 19 shows that few differences between the different variables in terms of stress, values, and location of stress can be observed consistently, so an impact on the variables temperature or speed is not seen in this case.

Figure 20 again shows punctual stress concentrations in areas far from rollers, and a trend towards the variables studied such as temperature or speed was not identified in the evaluated cases, nor towards the parameters recorded in the runs such as gradient or deformation speeds.

Figure 21 shows the temperature graph obtained for the fourth pass for all the different scenarios of this work. 


**Pass 5:**


Figure 22 shows the concentration of stress according to what was expected in magnitude and location.

In Figure 23 show the effective stress due to the simulations of the 5th pass of the upper rail and, unlike the other simulations, here small concentrations in different parts of the rail depending on the run can be seen, appearing in the upper part (1150 °C-12.4 RPM) or lower part (1150 °C-6.2 RPM) depending on the observed case.

In the results shown in Figure 24, similar behaviors are observed in terms of temperature, with consistent results in the nine runs and with minimal differences.

## 5. Conclusions

It was possible to simulate five hot rolling passes of the rail. More work needs to be developed to obtain more accurate results related to rolling load verification and validation. These simulations allow us to identify the study cases where future work must be deepened, detailing and segmenting the pass to identify other variables that explain that behavior.

Through the present work, it was possible to propose a methodology for the evaluation of rolling passes with different conditions of temperature and speed. The iterations allow better understanding of material flow with the purpose of saving costs, identifying areas of opportunity, and being able to optimize or improve the manufacturing process from process simulations.

An in-depth analysis of each pass using segmentations and considering other variables such as the insertion angle needs to be carried out. This is proposed as future work.

It is recommended to conduct simulations of each rolling with the temperature closest to the real process to determine the causes of failures. A undesired flow of material can result in product becoming stuck in situ in the rolling at the time of being processed, and thus it may be possible to improve the process and prevent this before it occurs.

By performing simulations, failures and improvements in the process can be determined, which saves raw materials and a lot of rework time.

Considering that the present study does not contemplate a real evaluation for physical validation, a comparison could be made with the work shown in [28] where work is described with a piece of similar dimensions and a temperature of 1200 °C, and the stress values reported in the publication oscillate between 130 MPa and 365 MPa, while the results of the present work show maximum values of stress of 295 MPa. Comparing the maximum value of the simulation, we can affirm that the results of the simulations are within a similar value range. It is also worth mentioning that [28] does not define fine data such as the % reduction of the pieces for the reported passes, and this information would allow us to make a detailed selection of the most appropriate pass for validation and comparison between the passes of the two publications.

## Figures and Tables

**Figure 1 materials-16-00002-f001:**
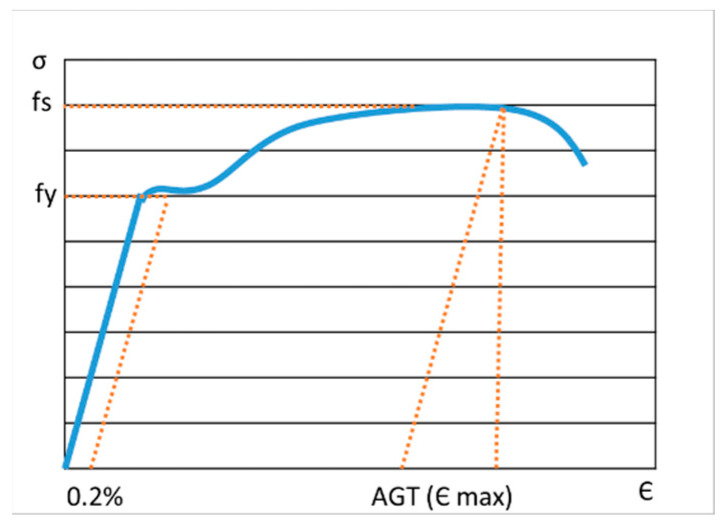
Stress–strain curve. fs: Ultimate strength, fy: Yield strength.

**Figure 2 materials-16-00002-f002:**
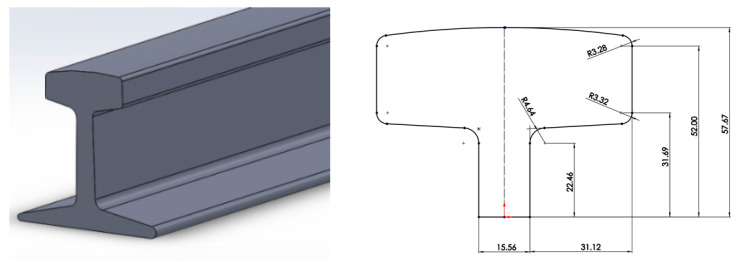
Complete railway and top cross section of rail in two equal segments.

**Figure 3 materials-16-00002-f003:**
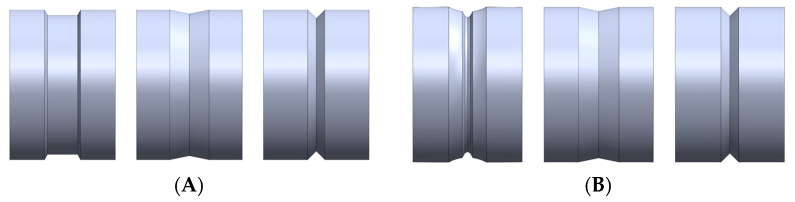
Bottom and top roll geometries for first 3 passes. (**A**) Bottom roll geometry. (**B**) Top roll geometry.

**Figure 4 materials-16-00002-f004:**
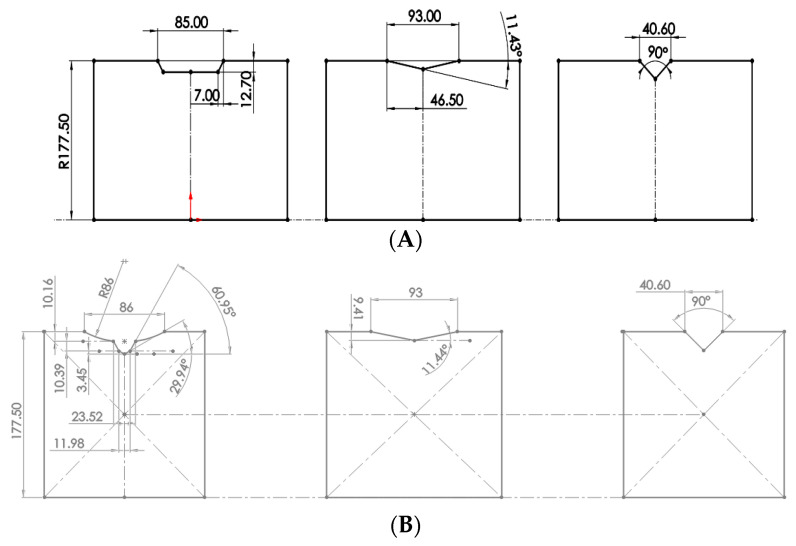
Roll dimensions (in mm). (**A**) Bottom roll dimensions (in mm). (**B**) Top roll dimensions (in mm).

**Figure 5 materials-16-00002-f005:**
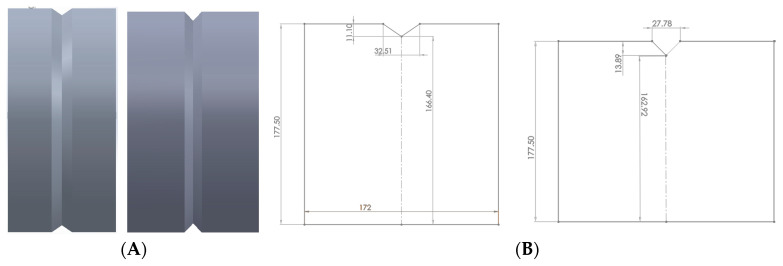
Geometries and dimensions for 4th and 5th passes. (**A**) The 4th and 5th geometry passes. (**B**) The 4th and 5th roll dimensions (in mm).

**Figure 6 materials-16-00002-f006:**
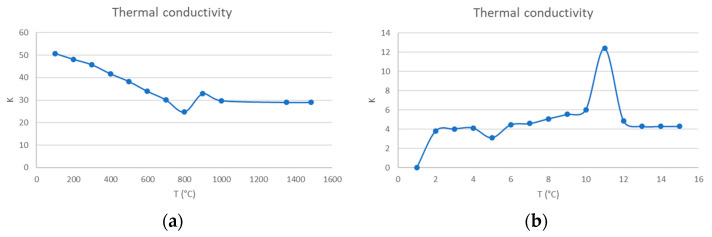
(**a**) Thermal conductivity of materials. (**b**) Heat capacity defined in the process.

**Figure 7 materials-16-00002-f007:**
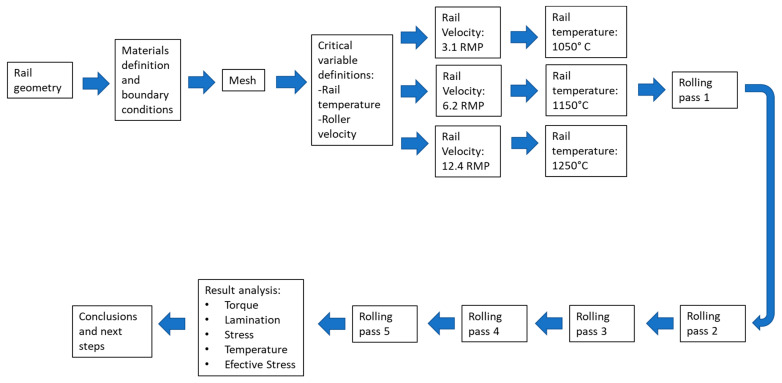
Upper rail experimentation flowchart.

**Figure 8 materials-16-00002-f008:**
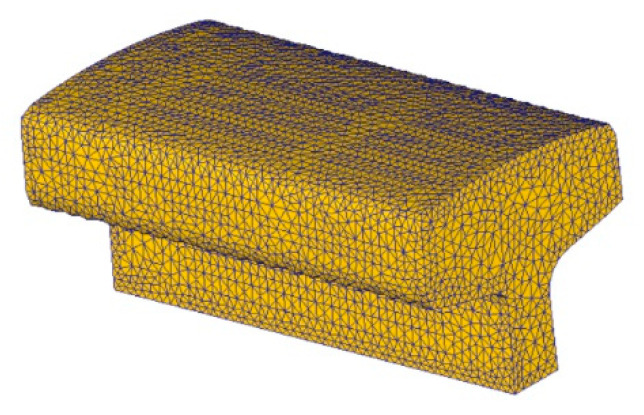
Upper segment of railway rail.

**Figure 9 materials-16-00002-f009:**
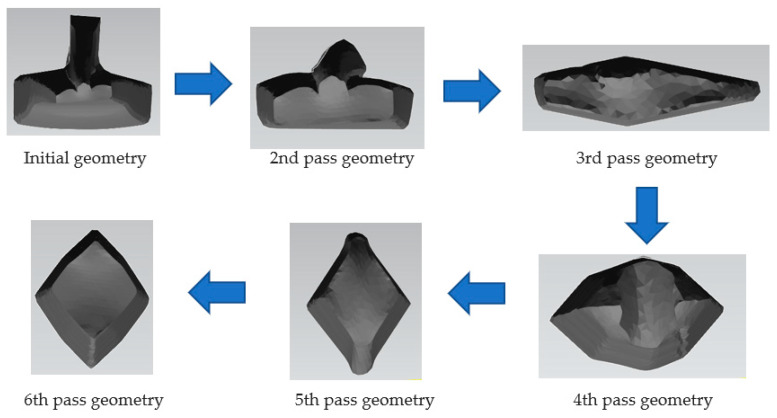
Railway segment for each of five roller passes before and after process.

**Figure 10 materials-16-00002-f010:**
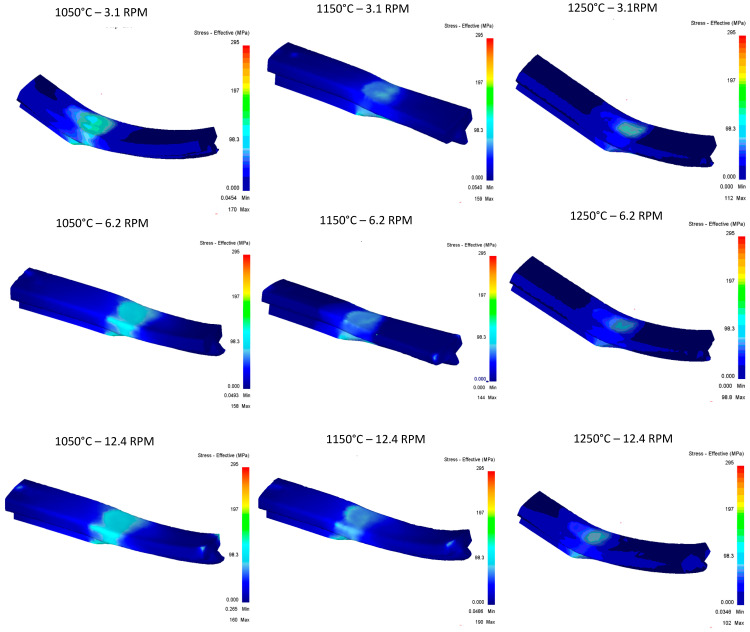
Effective stress in the top Rail after first pass.

**Figure 11 materials-16-00002-f011:**
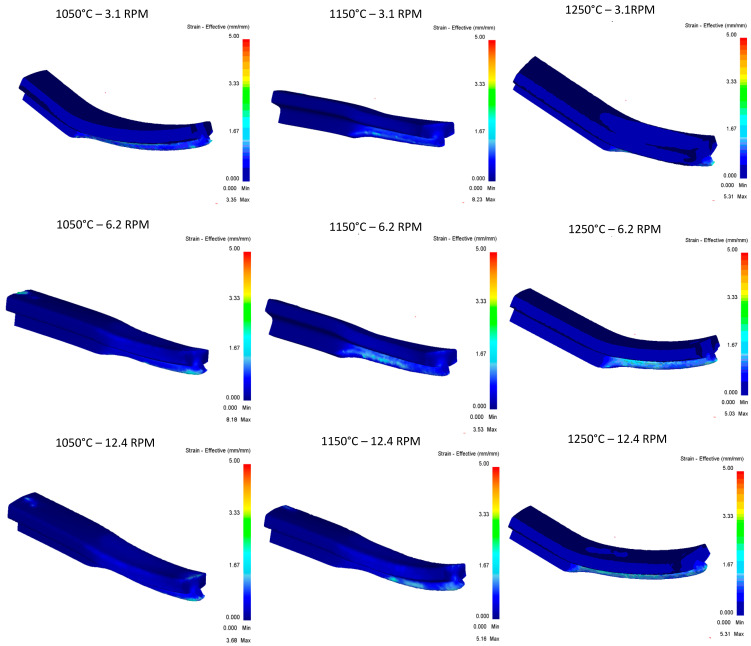
Effective strain in the top rail after first pass.

**Figure 12 materials-16-00002-f012:**
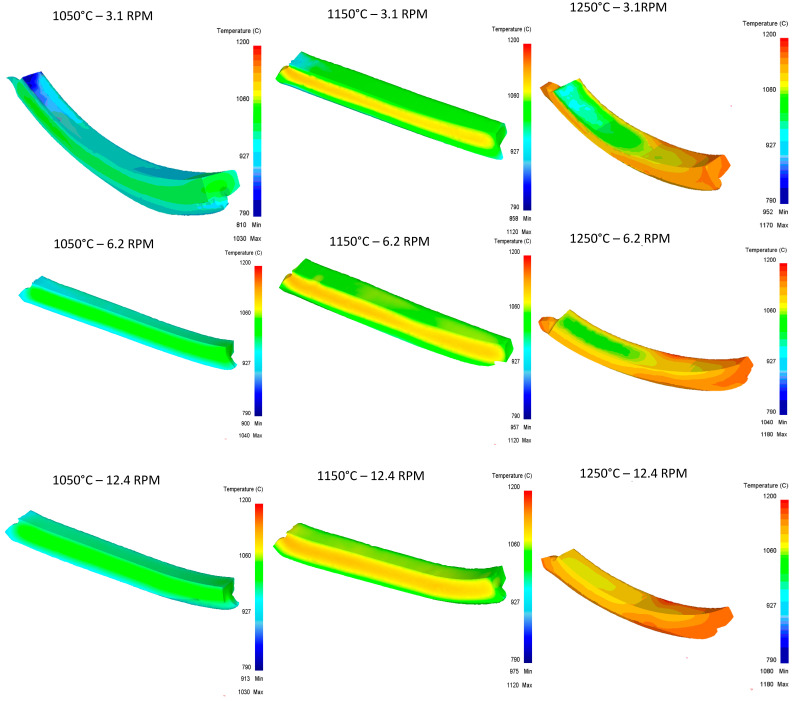
Upper Rail temperature graph for first pass.

**Figure 13 materials-16-00002-f013:**
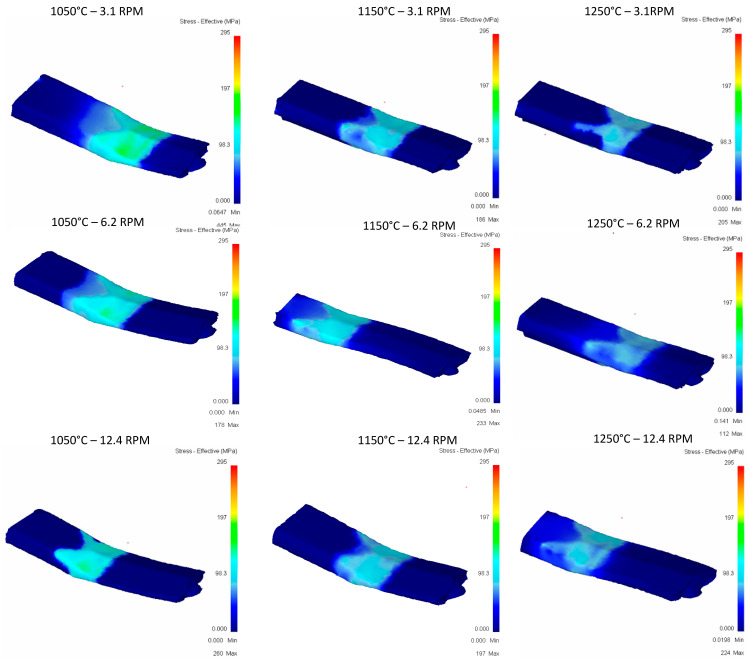
Effective stress in the top Rail after second pass.

**Figure 14 materials-16-00002-f014:**
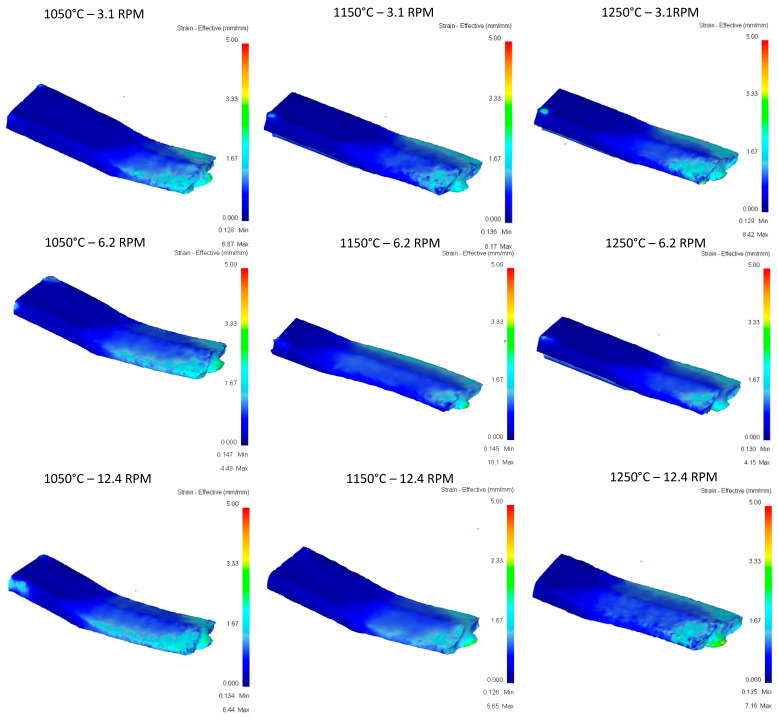
Effective Stress in the top Rail after second pass.

**Figure 15 materials-16-00002-f015:**
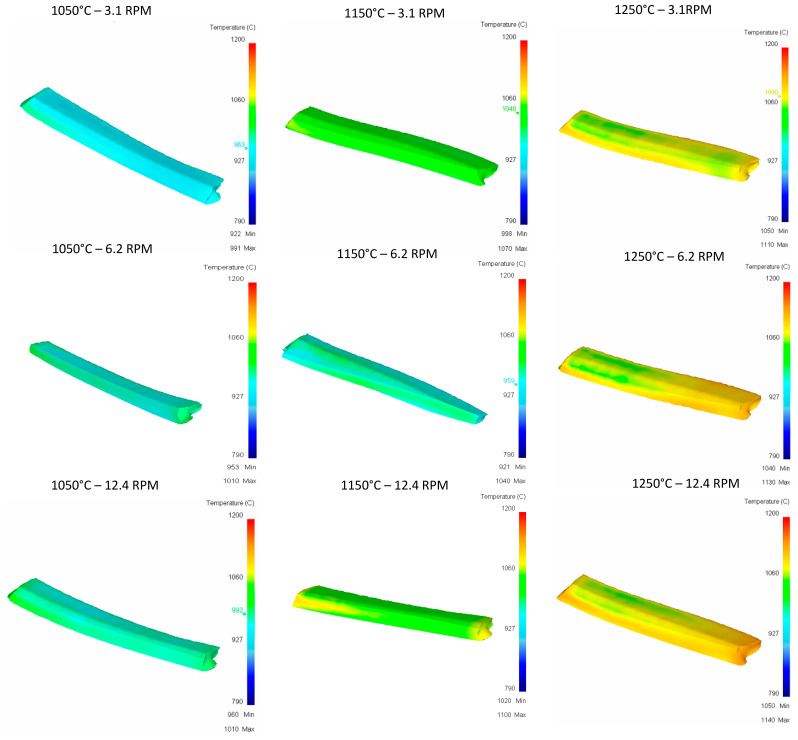
Upper Rail temperature graph for second pass.

**Figure 16 materials-16-00002-f016:**
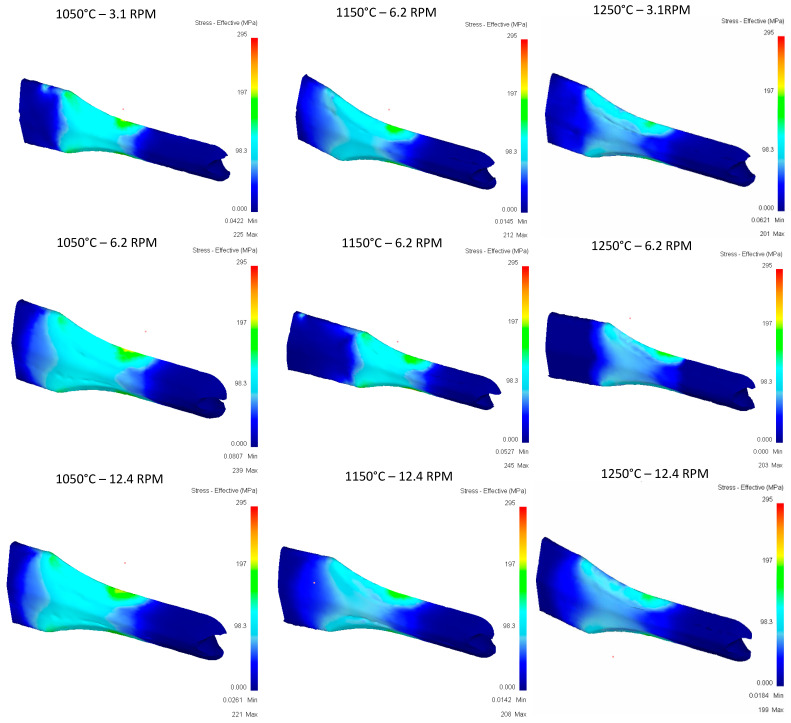
Effective stress in the top Rail after third pass.

**Figure 17 materials-16-00002-f017:**
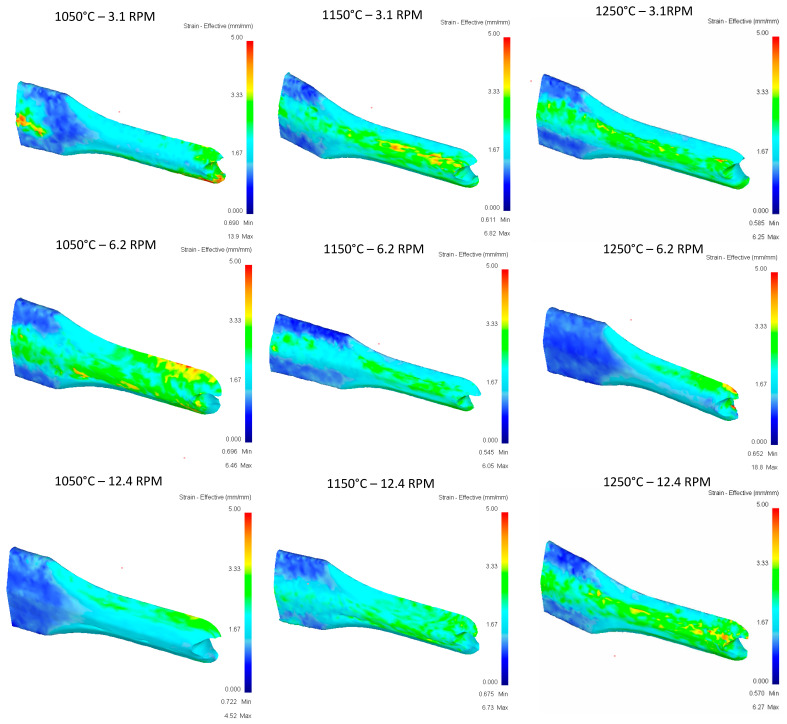
Effective Stress in the top Rail after third pass.

**Figure 18 materials-16-00002-f018:**
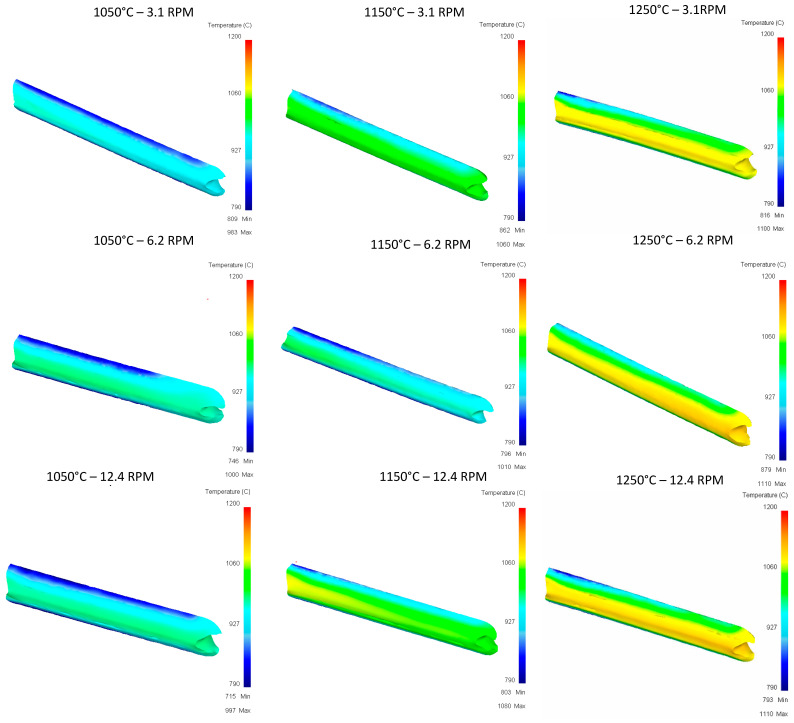
Upper Rail temperature graph for third pass.

**Figure 19 materials-16-00002-f019:**
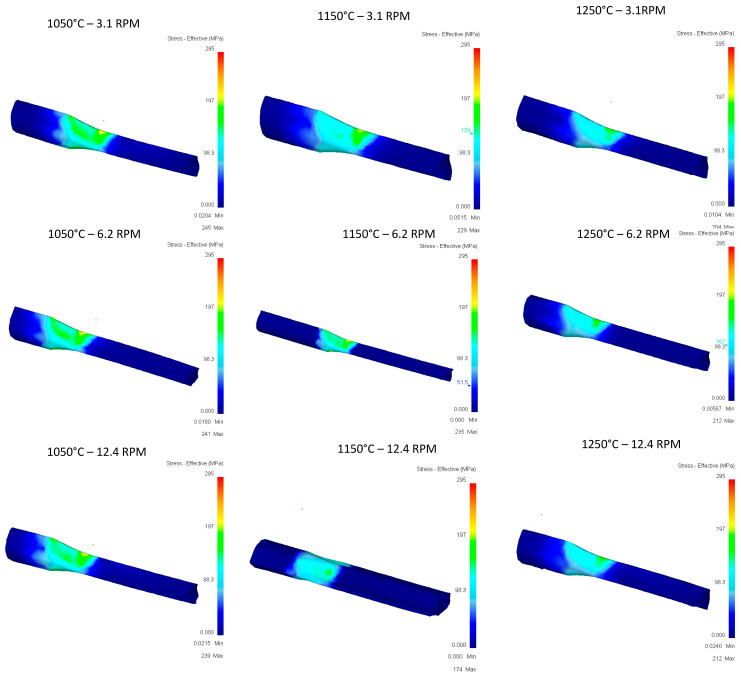
Effective stress in the top Rail after fourth pass.

**Figure 20 materials-16-00002-f020:**
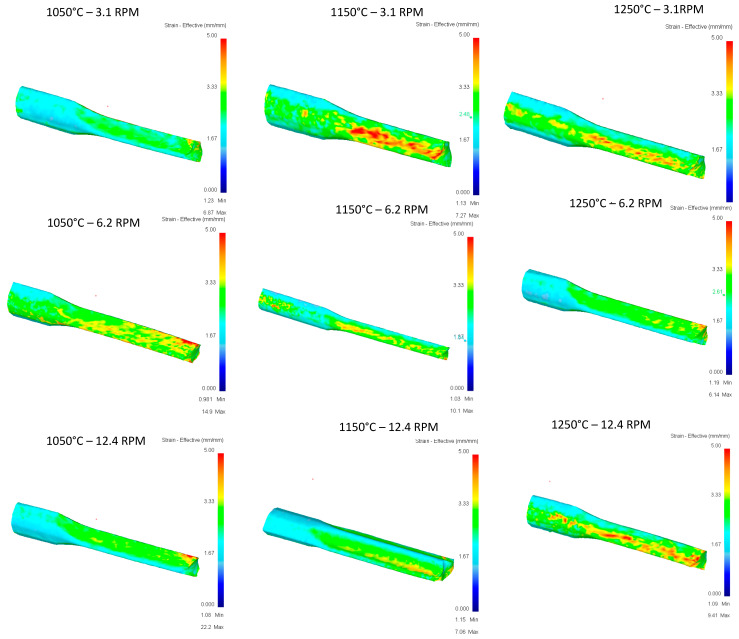
Effective Stress in the top Rail after fourth pass.

**Figure 21 materials-16-00002-f021:**
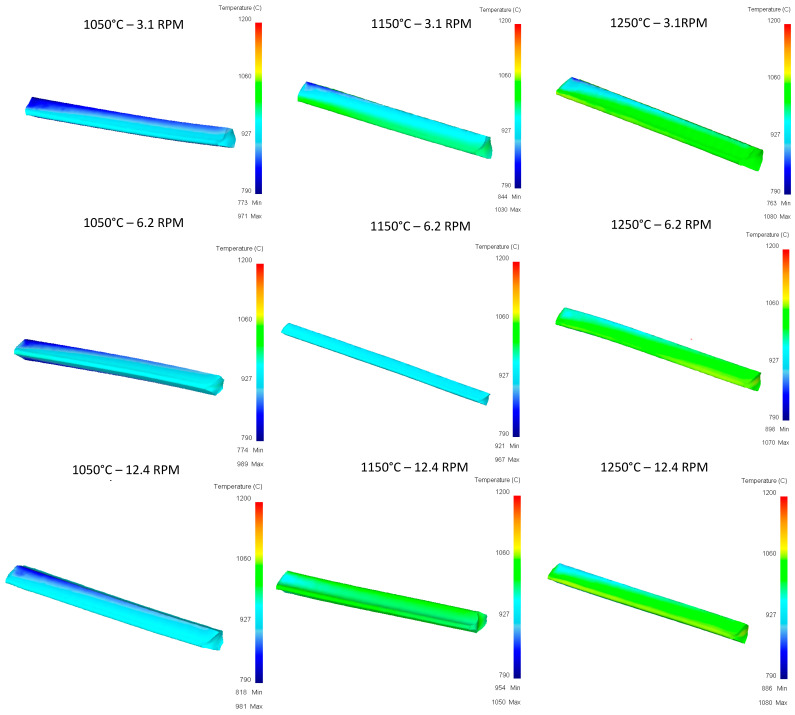
Upper Rail temperature graph for fourth pass.

**Figure 22 materials-16-00002-f022:**
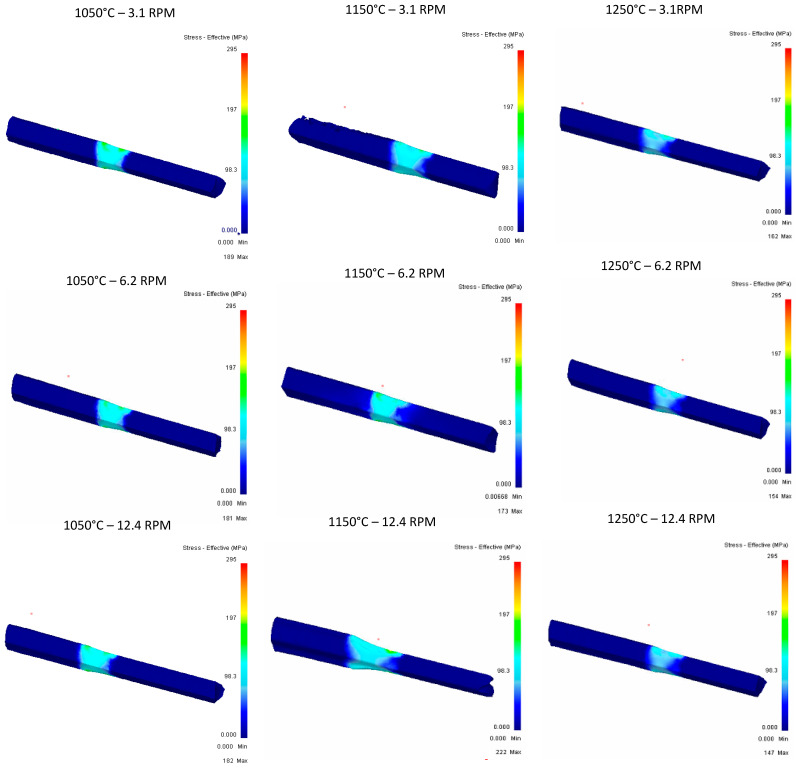
Effective stress in the top Rail after fifth pass.

**Figure 23 materials-16-00002-f023:**
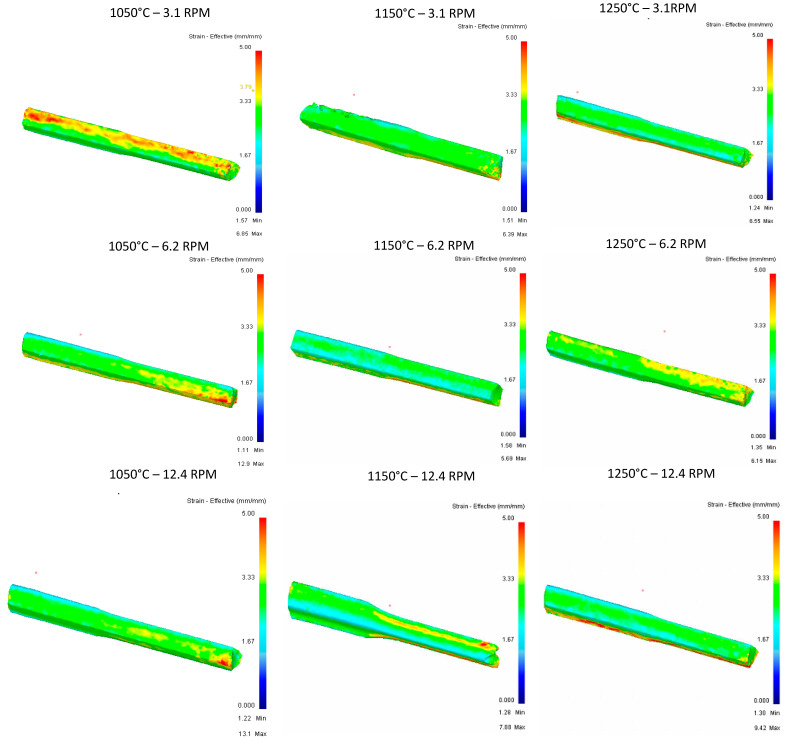
Effective Stress in the top Rail after fifth pass.

**Figure 24 materials-16-00002-f024:**
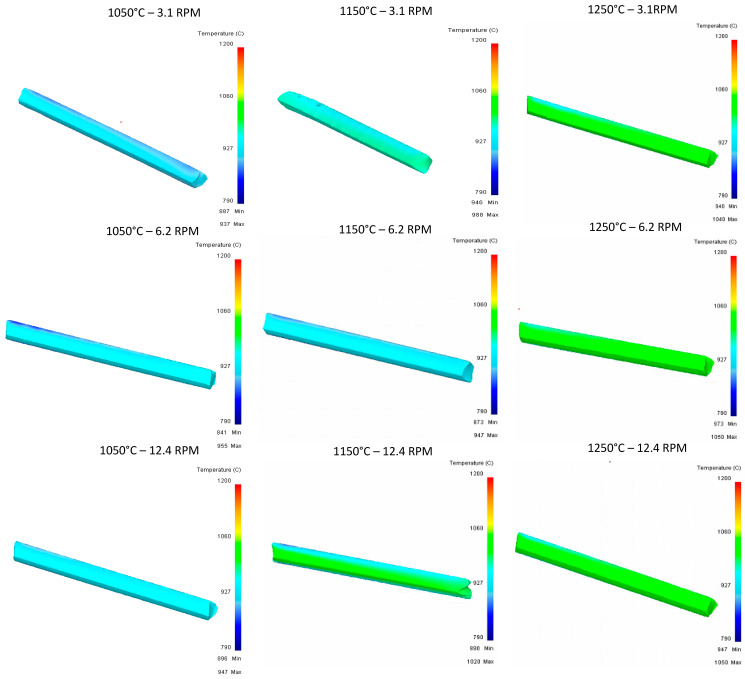
Upper Rail temperature graph for fifth pass.

**Table 1 materials-16-00002-t001:** Steel 1075 chemical composition.

Element	Percentage
C	0.7–0.8
Mn	0.4–0.7
S	<0.05
P	<0.04

**Table 2 materials-16-00002-t002:** Elastic properties of 1075 steel.

Young’s modulus (GPa)	206
Poisson’s ratio	0.3
Thermal expansion coefficient (°C^−1^)	1.2 × 10^−5^
Density (kg/m^3^)	7800

**Table 3 materials-16-00002-t003:** Basic information about DEFORM simulations for each pass.

Variable		Pass 0	Pass 1	Pass 2	Pass 3	Pass 4	Pass 5
Railway rail dimensions	Length (mm)	348.63	417.41	662.58	1001.09	1665.32	1829.52
	Width (mm)	70.39	73.05	82.03	44.85	27.17	26.9
	Height (mm)	57.59	41.75	24.07	29.4	37.92	32.67
Top roll diameter (mm)		-	355	355	355	304.8	304.8
Bottom roll diameter (mm)		-	355	355	355	304.8	304.8
Average value of deformation (reduction)		-	16.5%	32.5%	32.5%	36.7%	18.7%
Deformation velocity (m/s)	version 3.1 RPM	-	0.07	0.09	0.09	0.08	0.06
	version 6.2 RPM	-	0.14	0.17	0.17	0.16	0.12
	version 12.4 RPM	-	0.28	0.34	0.34	0.31	0.24
Machine time (s)		-	13.4	26.2	37.9	52	65.8
Contact area upper roller vs. pass mm^2^		-	1846	1655.8	233.1	91.6	61.9
Contact area lower roller vs. pass mm^2^		-	334	935	242	89	61.9

**Table 4 materials-16-00002-t004:** Rolling loads and torques for each pass.

	**Rolling Load (KN)**								
**T (°C)**	**1050**			**1150**			**1250**		
**Pass\** **Ang vel**	**3.1**	**6.2**	**12.4**	**3.1**	**6.2**	**12.4**	**3.1**	**6.2**	**12.4**
1	200	205	215	145	155	157	130	145	151
2	500	450	440	370	350	340	310	302	290
3	190	170	175	150	175	135	125	115	120
4	220	205	208	155	200	180	150	148	146
5	58	57	58	49	52	69	40	39	39
	**Rolling Torque (KN)**								
**T (°C)**	**1050**			**1150**			**1250**		
**Pass\** **Ang vel**	**3.1**	**6.2**	**12.4**	**3.1**	**6.2**	**12.4**	**3.1**	**6.2**	**12.4**
1	14	10	11	7.0	6.8	7.0	9.5	9.3	9.7
2	16	14.5	12.9	11.5	11.9	11	10	9.2	9.8
3	7.7	7.0	6.9	6.5	7.0	4.9	5.5	4.5	5.5
4	5	5.1	4.9	4.2	5.0	3.3	4.1	3.2	3.5
5	0.8	0.9	1.0	0.7	0.8	1.6	0.7	0.6	0.7

## Data Availability

Not applicable.

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
