# Peer review of "Finite Element Modeling of Hot Rolling of 1075 Carbon Steel Process with Variable Cross Section"

_materials, 2022, doi:10.3390/ma16010002_

Round 1
Reviewer 1 Report
This is timely effort conducted by authors on "Finite element modeling of hot rolling of 1075 carbon steel process with variable section". However, various portions of this manuscript need serious improvements.
1. Novelty seems to be little.
2. Useless repetitions of blocks in the Figure 4.
3. Overall, various grammatical and spelling mistakes e.g. Pase instead of pass. English should be written as per the good English standard for writing with clarity.
4. Please clearly writed the purpose of Figs. 6 and 7. And which of these figures, you are taking for current simulations.
5. Simulation results are not clearly presented and explained for each pass.
6. Too much figures, you can make a supplementary file to put extra figures there if they cant be skipped.
7. Abstract must have some glimpses of crucial results.
8. Conclusion was started very badly. It should be strongly improved.
9. What does it mean by thesis in this manuscript? Please the 3rd Experimentation heading and in first line.
10. Simulations are only reliable if they are validated! Why did you not validate these simulations?
11. If simulations are extremely expensive to be validated then their result may be compared with the literature to measure its reliability.
12. Figures must come later after text where they have been specified. In this manuscript, the reverse is true.
Thanks
Reviewer 2 Report
This paper presents a simulation and comparative analysis of the rolling process of rails at different temperatures and rolling speeds with the help of deform software. The study has a large workload, but is not innovative enough. Unlike plate rolling, the temperature difference between different parts of the rolling process is relatively large, and if the effect of the previous pass on the next pass is not taken into account, relatively large errors will occur in the simulation. In addition, the article does not verify the accuracy of the simulation results by experimental comparison.
Reviewer 3 Report
1. No chemical composition of the rolled steel.
2. No assembly drawing of the rolls with cross-sections of the rolled band, or separate drawings of individual roll blanks.
3. Lack of a table showing the basic data for modeling, including the initial dimensions of the feedstock: length, width, height (before rolling), the nominal diameters of the rolls, the working diameters of the rolls for each pass, the cross-sectional dimensions of the band in each pass, the average values of deformation and deformation velocity in each pass, the average linear velocity of the band in each pass, the residence time of the band in the pass (so-called machine time), the contact area of the band with the top roll and the bottom roll for each pass.
4. No information on the finite element mesh used for the deformed metal and rollers.
5. There are incorrect designations on page 4.Deformation is usually denoted as ε, strain rate as Î, and E is the longitudinal modulus of elasticity (Young's modulus).
6. Fig. 4 lacks data on the changes in the speed of the rollers, while changes in the temperature of the charge are given twice.
7. In Figure 1, there is no explanation of the designations fy, fs.
8. No analysis of the curves in Figure 18B, with very variable (fluctuating) values of moments. Is this due to uneven gripping of the band and sliding of the band on the rollers until the rolling process is established, or are these modeling errors?
9. On page 8 there is no indication of "Pass one".
10. The analysis of the data in Figure 10 - 29 is poor.
11. What was the purpose of the research conducted? After all, in the process of rolling rails, the temperature of the strip changes before each pass. The rolling process includes, in addition to the machine time(rolling), also the break time needed for auxiliary operations. The cross-sectional area of the strip is relatively small, so in the gaps between the "pure" rolling process there is a decrease in the temperature of the strip, which the authors did not take into account.
12.There is also a change in the value of flow stress, which directly affects the force-energy parameters of the rolling process.
13.The final conclusions are inadequate to the content of the article.
Round 2
Reviewer 1 Report
Authors have tried to comply with what my most of the comments, however, authors must observe rigorous improvements for following points:
1. Overall, various grammatical and spelling mistakes e.g., Pase instead of pass. English should be written as per the good English standard for writing with clarity.
2. Too many figures, you can make a supplementary file to put extra figures there if they can't be skipped.
3. Simulations are only reliable if they are validated! Why did you not validate these simulations?
4. If simulations are extremely expensive to be validated, then their result may be compared with the literature to measure its reliability.
5. Figures must come later after text where they have been specified. In this manuscript, the reverse is true.
Thanks
Author Response
Good morning, I attach the corrected document
Regards

Reviewer 3 Report
The comments and questions contained in the review have been included in the text of the article.
Author Response

(The authors gave the same response as above.)
